# Impact of Running Clothes on Accuracy of Smartphone-Based 2D Joint Kinematic Assessment During Treadmill Running Using OpenPifPaf

**DOI:** 10.3390/s25030934

**Published:** 2025-02-04

**Authors:** Nicolas Lambricht, Alexandre Englebert, Anh Phong Nguyen, Paul Fisette, Laurent Pitance, Christine Detrembleur

**Affiliations:** 1Institute of Experimental and Clinical Research, UCLouvain, 1200 Brussels, Belgium; nicolas.lambricht@uclouvain.be (N.L.); anhphong.nguyen@uclouvain.be (A.P.N.); laurent.pitance@uclouvain.be (L.P.); 2Institute of Information and Communication Technologies, Electronic and Applied Mathematics, UCLouvain, 1348 Louvain-la-Neuve, Belgium; alexandre.englebert@uclouvain.be; 3Institute of Mechanics, Materials and Civil Engineering, UCLouvain, 1348 Louvain-la-Neuve, Belgium; paul.fisette@uclouvain.be; 4Service de Stomatologie et de Chirurgie Maxillo-Faciale, Cliniques Universitaires Saint-Luc, 1200 Brussels, Belgium

**Keywords:** OpenPifPaf, running, clothing, 2D motion analysis, markerless, human pose detection

## Abstract

The assessment of running kinematics is essential for injury prevention and rehabilitation, including anterior cruciate ligament sprains. Recent advances in computer vision have enabled the development of tools for quantifying kinematics in research and clinical settings. This study evaluated the accuracy of an OpenPifPaf-based markerless method for assessing sagittal plane kinematics of the ankle, knee, and hip during treadmill running using smartphone video footage and examined the impact of clothing on the results. Thirty healthy participants ran at 2.5 and 3.6 m/s under two conditions: (1) wearing minimal clothing with markers to record kinematics by using both a smartphone and a marker-based system, and (2) wearing usual running clothes and recording kinematics by only using a smartphone. Joint angles, averaged over 20 cycles, were analysed using SPM1D and RMSE. The markerless method produced kinematic waveforms closely matching the marker-based results, with RMSEs of 5.6° (hip), 3.5° (ankle), and 2.9° (knee), despite some significant differences identified by SPM1D. Clothing had minimal impact, with RMSEs under 2.8° for all joints. These findings highlight the potential of the OpenPifPaf-based markerless method as an accessible, simple, and reliable tool for assessing running kinematics, even in natural attire, for research and clinical applications.

## 1. Introduction

Running is a popular sporting activity, primarily due to its accessibility and the associated health and longevity benefits [1]. Despite these benefits, the incidence of overuse injuries to the lower limbs remains high among novice, recreational, and competitive runners [2]. Indeed, this activity involves the high repetition of rapid unipodal movements, exposing participants to significant external forces which can be several times their body weight [3]. Running technique is considered a risk factor for running injuries [4], making the evaluation of kinematics valuable not only during training but also in the rehabilitation of injured runners and other athletes.

In the particular case of anterior cruciate ligament (ACL) rupture, a common injury in young and active individuals [5], return to running is an important milestone in rehabilitation [6,7]. A systematic review has highlighted significant differences in knee joint kinematics between ACL-injured patients and patients with uninjured legs and in control groups during running [8]. The authors noted that alterations particularly affect the sagittal plane, resulting in altered knee kinematics, and persisted for up to five years post-reconstruction. In addition, altered movement patterns have been implicated in the early onset of knee osteoarthritis [9] and in increased risks of re-injury [10]. Therefore, a comprehensive assessment of knee kinematics and a focus on restoring running biomechanics early in rehabilitation may be beneficial for patients [11].

Kinematic parameters are usually assessed using an optoelectronic system consisting of multiple infra-red cameras and reflective markers placed on the subjects. However, the high cost, need for dedicated space and skilled operators, and lack of portability make these systems difficult to use outside research settings [12]. To address these limitations, various markerless technologies have been investigated. Inertial measurement units (IMUs) [13], depth cameras (e.g., the Kinect) [14], and single or networked RGB cameras have been extensively studied [15,16]. For the third approach, the development of deep learning techniques for human pose detection has enabled markerless computation of 2D and 3D motion kinematics from single or multiple 2D video sources [17,18]. In particular, the kinematics in the sagittal plane derived from various pose estimation methods have been validated against traditional 2D tools, such as Kinovea, as well as reference 3D motion analysis systems during walking, running, and tasks like squatting or jumping [19,20,21]. When analysing running kinematics in the sagittal plane, two approaches can be employed: measuring 3D kinematics and extracting sagittal plane data, or directly measuring only the sagittal plane in 2D. For the first approach, multiple cameras are required, which must be synchronized and calibrated. Traditionally, the 2D coordinates of anatomical key points are extracted from each video using neural networks, such as OpenPose [22]. Subsequently, 3D coordinates are calculated by triangulating the measurements from the different cameras, and kinematics are derived using inverse kinematics [15,23]. The second approach, while less robust and versatile, is more accessible and easier to implement in clinical practice, as it requires only a single camera, such as a smartphone camera. Researchers have evaluated the performance of OpenPose-based systems for kinematics evaluation [20], as well as other solutions requiring lower computing costs, such as BlazePose [16]. OpenPifPaf [24,25] is a convolutional neural network designed for human pose detection that shows good performance, outperforming OpenPose in challenging conditions, such as low-resolution and crowded, cluttered, or occluded scenes [26]. A recent study compared sagittal plane kinematics during jumps measured based on OpenPifPaf to those obtained using a marker-based system [27]. The results highlighted its potential clinical value due to its performance and ease of use. However, its application for running analysis has not yet been explored. Unlike marker-based systems, markerless motion capture does not require attaching markers to body segments. Instead, it relies on automatic landmark estimation, which can be influenced by factors such as clothing [28]. Recent studies indicate that clothing has minimal impact on 3D markerless walking and running kinematics measured with multiple cameras [29,30,31]. Investigating whether this holds for single-camera 2D sagittal plane studies would be valuable.

The primary aim of this study was to evaluate the performance of an OpenPifPaf-based markerless method for measuring running kinematics. This involved comparing its results for hip, knee, and ankle joints in the sagittal plane with those obtained using a 2D marker-based approach. The secondary aim was to examine the effect of subjects wearing their usual running clothes on the markerless method results.

## 2. Materials and Methods

### 2.1. Participants

The participants were 30 healthy men and women (15/15; age: 23.1 ± 4.0 years old; height: 1.77 ± 0.1 m; mass: 66.6 ± 9.1 kg) who had not suffered any lower extremity musculoskeletal injuries for at least 6 months and were able to run at 3.6 ms^−1^ for at least 3 min. Most of the participants were young, Caucasian, and in good physical condition.

This study was approved by the ethics committee of our university (B403201523492). The purpose of the study was explained to the participants and they all provided their written informed consent to the use of their anonymized data before the experiments.

### 2.2. Procedure

Marker-based motion capture was conducted at 100 Hz using a height-camera optoelectronic system (Vicon V5 Motion Systems, Oxford Metrics Ltd., Oxford, UK) positioned around the treadmill. For markerless analysis, a smartphone (iPhone12; frequency: 60 Hz; resolution: 1280 × 720 pixels) was mounted on a tripod with a fixed height of 1 m and placed 3 m laterally perpendicular to the movement plane (Figure 1A).

During the experiment, movement was recorded for 30 s as participants ran at two fixed speeds, 2.5 m.s^−1^ and 3.6 m.s^−1^, under two clothing conditions. The first condition (‘Lab clothing’) required participants to wear close-fitting underwear, their running shoes, and eighteen infra-red reflective markers. Sixteen markers were used to comply with the Nexus ‘Plug-in Gait lower body’ model, with an additional marker placed on the middle of the lateral edge of each acromion. In the second condition (‘Running clothes’), participants wore their usual running clothes without any specific guidance, typically shorts or leggings for the bottoms and loose-fitting t-shirts for the tops.

Participants were given a 5-min familiarization period to adjust to treadmill running and select a comfortable speed. During this time, they were instructed to test and briefly run at the two experimental speeds to ensure familiarity with these conditions. After the familiarization period, participants were equipped with the reflective markers and ran on the treadmill at the lower speed. After 60 s, they were asked whether they felt comfortable at this speed. Once participants indicated they were comfortable, data collection was initiated, capturing 30 s of motion. The same process was then repeated for the higher speed after a 2-min pause between the two speeds. Between the two clothing conditions, a longer pause was required to allow for the removal of motion capture markers and for participants to dress. Although the exact duration of this pause was not recorded, care was taken to ensure participants were ready before resuming the protocol. In both clothing conditions, participants always started with the lower speed.

Under the Lab clothing condition, a smartphone video was recorded simultaneously with the static calibration of the optoelectronic system prior to capturing running motion. Motion under this condition was captured using both the optoelectronic system and the smartphone. To synchronize the recordings and allow for the selection of the same running cycles during analysis, participants performed a ‘butt-kick’ movement approximately one second after the start of motion capture recording. Under the Running clothes condition, movements were recorded exclusively via smartphone video, and no ‘butt-kick’ trigger movement was required.

### 2.3. Data Processing

Spyder scientific environment (Python 3.12) was used for data processing. Since the smartphone was positioned to the left of the participants, only the joint kinematics on that side were assessed. For the marker-based reference system, kinematics were not computed using the Plug-in-Gait model. Instead, the optoelectronic system was used to extract accurate spatial coordinates of the markers placed on anatomical landmarks, similar to the key points identified by OpenPifPaf, for joint kinematics computation. The raw 3D marker trajectory data from the optoelectronic system were exported and filtered using a low-pass, zero-lag, fourth-order Butterworth filter with a cut-off frequency of 6 Hz. The approximate position of the left hip joint was estimated using the Newington–Gage model [32]. This estimation involved utilizing data from the four superior iliac spine markers as well as the participant’s leg length. The 2D coordinates of the four markers on the participant’s left side, along with the hip position estimated by the Newington–Gage model (represented by triangles and a star in Figure 1), were used for kinematics computation. For the markerless pose detection, we used the WholeBody extension of OpenPifPaf with the shufflenetv2k30-wholebody checkpoint. The video was processed by the neural network after rescaling by a factor 0.5 to 640 × 360 pixels. This allowed us to obtain the coordinates of five key points of interest for joint kinematics computation (represented by black dots in Figure 1). Key point trajectory gaps were filled using cubic spline interpolation, and the data were filtered using the same low-pass Butterworth filter as applied to the optoelectronic system data. Joint kinematics were calculated similarly for both methods using trigonometry and the dot product. For the ankle, the foot was considered a rigid body, and joint angles were corrected by accounting for the average angles measured in the neutral position during static calibration. The markerless method used key points of the knee, ankle, and big toe, while the marker-based method used markers on the lateral epicondyle, lateral malleolus, and second metatarsal. For the knee, the markerless method used key points of the hip, knee, and ankle, while the marker-based method used markers on the lateral malleolus and lateral epicondyle, and the estimated hip position. For the hip, the markerless method used key points of the shoulder, hip, and knee, while the marker-based method used markers on the acromion and lateral epicondyle, and the estimated hip location.

Figure 2 provides an overview of the procedure used to obtain the average joint kinematic curves over 20 cycles for the ankle, knee, and hip of a participant at a given speed and under a specific condition. These average curves were calculated for each of the 30 participants at the two speeds and across the three conditions: marker-based, markerless in lab clothing, and markerless in running clothes. For the Lab clothing condition, to ensure consistency in comparing the results obtained with the marker-based system and without markers, we utilized the ‘butt-kick’ movement as a trigger to select corresponding cycles. Indeed, during this trigger movement, the left ankle reached an abnormally high vertical position compared to the rest of the recording (see Trigger in Figure 2A). Then, we defined the start and end indexes of each running cycle based on the horizontal position of the left ankle. Given the treadmill setting, we inferred that the foot’s ground contact aligned with the ankle reaching its maximum anterior position. For subsequent analysis, we selected the indexes of the 20 cycles immediately following the second cycle after the trigger movement (see Figure 2A). The 2D joint kinematics for the ankle, knee, and hip joints based on both the markers and key point coordinates were computed (see Figure 2B). Then, the joint angles were segmented into running cycles using the previously determined indexes. These 20 cycles were resampled using linear interpolation to represent joint angles as a percentage of the cycle. Finally, the average angle for each joint was calculated at each percentage point across the 20 cycles (see Figure 2C). These averaged angles were subsequently used for further analysis.

A similar procedure was used to obtain average angle curves when running with usual clothes. Since the movement was recorded using only a smartphone, no trigger movement was required, and the twenty running cycles following the second cycle were used for analysis.

### 2.4. Statistical Analysis

To evaluate the performance of the markerless method, the averaged angle curves for the ankle, knee, and hip joints obtained without markers were compared to those obtained using markers at both running speeds. To assess how the differences between the angles measured by the two systems evolved throughout the running cycle, the absolute error was calculated at each percentage of the cycle for each participant and averaged across all 30 participants.

The Root Mean Square Error (RMSE) was calculated for the three joints for each participant. This metric provides an assessment of the overall accuracy of the markerless system in estimating lower limb kinematics and facilitates comparison with recent studies.

To analyse differences in joint kinematics between the two systems, time-series data for all participants were subjected to a one-dimensional Statistical Parametric Mapping two-way repeated-measures ANOVA using the SPM1D package (v0.4.22) [33]. The normality of the data was assessed prior to the analysis, and the alpha level of 0.05 was adjusted using Sidak correction to account for multiple comparisons, resulting in an effective alpha level of 0.017 for the three joints. The analysis included two factors: the system used (marker-based versus markerless) and the running speed. Although the ANOVA allows for examining three effects—the main effect of the system, the main effect of speed, and the interaction between system and speed—we focused exclusively on the main effect of the system and the interaction between system and speed, as these were most relevant to our research objectives.

To investigate the potential effect of clothing, the same methods were applied to compare joint kinematics obtained with the markerless system under Lab clothing and Running clothes conditions.

## 3. Results

Figure 3 presents the comparison between the markerless system results and those of the marker-based reference system throughout the running cycle. It includes the joint kinematic curves for the ankle, knee, and hip joints at 2.5 m.s^−1^ and 3.6 m.s^−1^, the results of the SPM1D analyses for system comparison and speed influence, and the mean absolute differences between the two systems. The SPM1D analysis revealed significant differences between the marker-based and markerless systems, with minimal influence of speed on these differences. The biggest discrepancy was observed at the hip joint, particularly between 20% and 60% of the cycle. Although we did not explicitly delimit the stance and swing phases during running in our study, this region approximately corresponds to the transition from the end of the stance phase to the beginning of the swing phase. Indeed, maximum hip extension typically occurs at the moment of toe-off during running [34]. During this period, the markerless system reported up to 9° greater hip extension compared to the marker-based system. Another notable difference occurred at the knee joint during the final 10% of the cycle, just before foot–ground contact, where the markerless system reported approximately 8° greater knee flexion.

Figure 4 compares the markerless kinematic results across the two clothing conditions. The joint kinematic curves appear very similar, and the SPM1D analysis comparing Lab clothing and Running clothes conditions reveals no significant differences. Additionally, the effect of speed on these results remains minimal. For all three joints, the maximum mean absolute error does not exceed 4°.

Figure 5 illustrates the distributions of the RMSEs for the three joints across the 30 participants at 2.5 m.s^−1^ and 3.6 m.s^−1^ for both systems. When comparing marker-based and markerless kinematics, the mean RMSE values range from 2.9° to 5.6°, depending on the joint and the speed. In contrast, the comparison of kinematic curves obtained by the markerless system under different clothing conditions yields mean RMSE values between 2.1° and 2.8°. Detailed RMSE values for all joints and conditions are provided in Table 1.

## 4. Discussion

In this study, we first compared the joint kinematics of the ankle, knee, and hip during running at 2.5 m.s^−1^ and 3.6 m.s^−1^, obtained using a 2D marker-based approach and an OpenPifPaf-based markerless 2D method. Overall, the results were consistent across both speeds. Despite several significant differences identified by the SPM1D analysis, the kinematic waveforms obtained without markers for the three joints compared reasonably well with those obtained from the marker-based system (see Figure 3). The averaged RMSE was highest for the hip joint at 5.6°, with lower values for the ankle and knee joints at 3.5° and 2.9°, respectively. Next, to assess the feasibility of testing participants in their usual running clothes instead of their underwear, we also investigated the influence of wearing running clothes on the markerless joint kinematic results. The curves obtained under both clothing conditions were highly similar, with no significant differences observed (see Figure 4), and the mean RMSE values were less than 2.8° for all joints. These findings are consistent with recent studies suggesting that clothing has minimal effect on 3D markerless walking and running kinematics measured using multiple cameras [29,30,31]. Furthermore, this indicates that the markerless method demonstrates acceptable repeatability [35] and robustness in consistently providing joint kinematics, even when participants wear their usual running clothes.

The accuracy of joint angle measurements during running cycles using OpenPifPaf is consistent with other studies that have compared pose detection methods to marker-based reference motion capture systems during gait and running. Focusing on gait, one study [36] reported mean absolute errors in the sagittal plane of 4 ± 2.1°, 5.6 ± 2.9°, and 7.4 ± 4.8° for the hip, knee, and ankle joints, respectively, using OpenPose for human pose detection. Another recent study [37] found RMSEs of 5.4 ± 2.1°, 5.7 ± 2°, and 4.6 ± 1.7° for the hip, knee, and ankle joint in the sagittal plane under normal walking conditions using OpenCap, a web-based service that utilizes videos from at least two smartphones to compute 3D movement dynamics [23]. Similarly, during running, a recent study [20] demonstrated mean RMSE values of 4.9 ± 2.2° for the hip, 6.5 ± 2.4° for the knee, and 5.7 ± 1.4° for the ankle when comparing OpenPose to an optoelectronic system, but only after temporal alignment and offset removal of their data. They also trained and tested a DeepLabCut model [38], which yields slightly higher RMSE values. This model was trained using video footage with reflective markers and was unable to detect the ankle and foot key points when videos of patients without reflective markers were tested. When comparing these results with other studies, it is important to consider that the method used to locate joint centres and the approach used to calculate joint kinematics can influence the outcomes. Additionally, the reference against which the results are compared also plays a significant role. For example, a study comparing two widely used musculoskeletal models, OpenSim and Plug-in-Gait, during running reported average RMSEs of 5.2°, 8.3°, and 5.5° for the hip, knee, and ankle at a running speed of 3.3 m.s^−1^ [39]. Therefore, a comparison of accuracy with other studies should be interpreted as an overview of their order of magnitude rather than as a basis for exact comparison. The lower RMSEs observed in our study are likely due to the fact that the comparison of the markerless method was made with a 2D method using markers fixed to the subjects, rather than with a 3D biomechanical model. This configuration is probably closer to the annotated pictures that the OpenPifPaf neural network was trained with. Furthermore, the same method was used to compute joint kinematics from the coordinates of the key points and markers for both the markerless and marker-based methods, ensuring a consistent approach to kinematic calculation.

From a clinical point of view, non-laboratory approaches for gait and running analysis, such as those relying on pose detection, should ideally capture joint kinematics in the sagittal plane with a maximum acceptable error of less than 5° [40,41]. The mean RMSE values for the knee and ankle angles fall below this threshold, indicating satisfactory performance. However, this is not the case for the hip, where the markerless method measures greater hip extension at toe-off, leading to a higher mean RMSE. This discrepancy might be explained by a mismatch between the spatial localisation of the markers on the acromion and the shoulder key point detected by OpenPifPaf. Additionally, at toe-off, the shoulders and pelvis are at their maximal rotation in opposite directions. With the limitation of 2D, both markers and key points are particularly susceptible to cross-talk artefacts [42], causing additional differences between the two systems. On the contrary, the RMSE for the hip is similar to that of the knee and ankle joints when comparing the markerless system with and without running clothes, highlighting the consistency of OpenPifPaf in detecting key points under both conditions. In the particular case of ACL injury, a systematic review reported alterations in knee flexion motion for the involved limb during the stance phase of running. These alterations were observed when compared with both the contralateral limb and control subjects, from 3 months to 5 years post-ACL reconstruction [8]. The authors emphasized the clinical significance of 2D video analysis in detecting and monitoring knee flexion excursion alterations during running. While traditional 2D motion analysis systems based on video offer an accessible tool to clinicians [43], pose estimation brings additional advantages. Firstly, it eliminates the tedious and time-consuming need to manually identify anatomical landmarks. Secondly, the pose estimation method provides the temporal evolution of the movement throughout its entirety, rather than being limited to discrete points of interest such as initial contact, toe-off, or peak knee flexion angle. These continuous curves enable not only the measurement and comparison of specific point values but also the evaluation of entire movement patterns, potentially allowing for the computation of more complex parameters involving temporal variations. Thirdly, using human pose detection for motion analysis when running on a treadmill allows for easy averaging over a large number of cycles, making the results more stable [44]. The use of a treadmill has also been shown to improve between-day repeatability of kinematic running assessment [45].

Our results should be interpreted within the context of our study’s limitations. First, we compared joint kinematics obtained from a markerless system with those obtained using markers placed on anatomical landmarks. This approach aligns with typical 2D motion analysis systems, such as Kinovea, but differs from the standard 3D motion analysis reference. In particular, the hip joint angle measured was between the thigh segment and the upper body segment, not between the pelvis and femur. Additionally, despite markers being placed by the same experienced operator, marker-based kinematics can be affected by palpation errors [46] and skin movement relative to the anatomical landmark [47]. Moreover, our method is limited to two-dimensional analysis exclusively in the sagittal plane, requiring perpendicular alignment of video recording to the treadmill. Deviations from this alignment could compromise measurement accuracy and relevance. Finally, our participants were healthy and predominantly Caucasian. Future studies should include a larger and more diverse sample, including actual patients, to broaden the applicability of our findings. Testing the method in real clinical conditions, outside the controlled laboratory setting, would be valuable. This would account for less standardized factors such as lighting and available space. Using a low-powered pose estimation framework [16] could enhance accessibility by reducing the need for specialized equipment and shortening computation time.

In summary, analysing treadmill running kinematics in the sagittal plane using OpenPifPaf for human pose detection yields compelling results compared to those obtained using markers placed on anatomical landmarks to simulate typical video-based 2D motion analysis systems. This method provides valuable insights into joint kinematics throughout the entire running cycle. The use of a treadmill and averaging results over a large number of cycles increase the method’s robustness. Additionally, wearing running clothes does not appear to significantly affect the method’s outcomes. Markerless motion capture shows great promise for both research and clinical practice due to its accessibility and simplicity, making it well suited for integration into clinical settings.

## Figures and Tables

**Figure 1 sensors-25-00934-f001:**
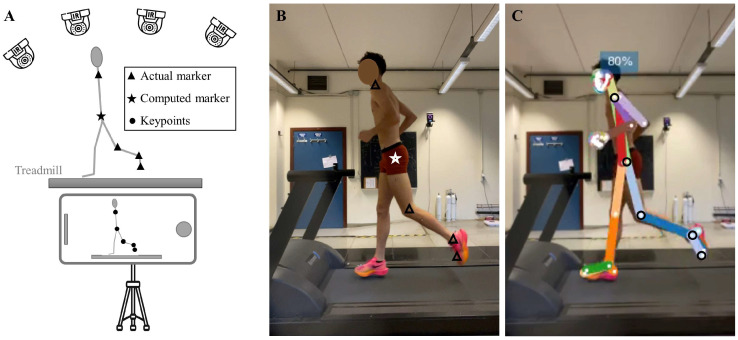
(**A**) A schematic of the experimental setup where motion is simultaneously recorded by an optoelectronic system and a smartphone camera. The images show a participant in the Lab clothing condition performing the experiment, both before (**B**) and after (**C**) OpenPifPaf processing.

**Figure 2 sensors-25-00934-f002:**
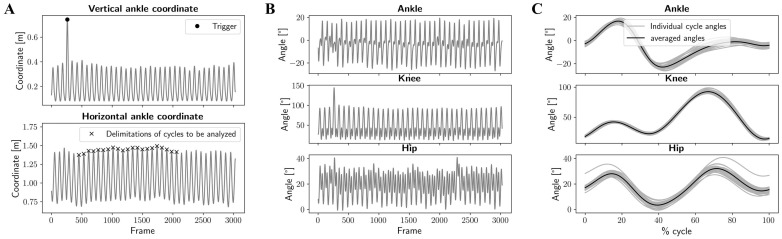
Examples of curves obtained for a participant at one speed. (**A**) The horizontal and vertical coordinates of the left ankle marker or key point, highlighting the synchronization trigger and the start and end indices of the cycles to be analysed. (**B**) The evolution of joint angles for the ankle, knee, and hip during the recording. (**C**) Curves showing joint angles at each cycle percentage for the 20 analysed cycles (grey) and their mean values (black).

**Figure 3 sensors-25-00934-f003:**
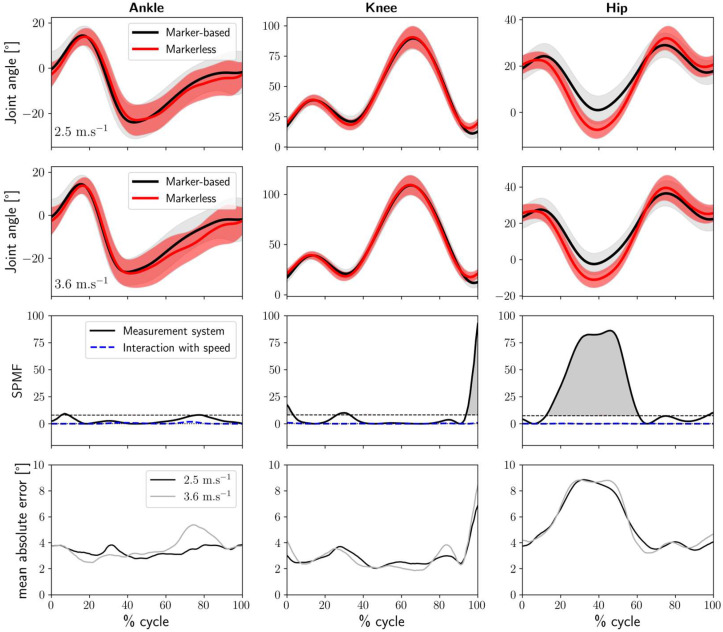
**Top**: Marker-based and markerless motion capture sagittal plane joint angles at 2.5 m.s^−1^ and 3.6 m.s^−1^ for the ankle (**left column**), knee (**middle column**), and hip (**right column**) across the 30 participants. The solid lines represent the mean of the averaged angles of the 30 participants, and the shaded areas represent the standard deviation. **Middle**: F-statistics from SPM1D (SPMF) for the measurement method (solid line) and the interaction of speed and measurement method (dotted line), with dashed lines indicating the critical thresholds and regions of significant differences highlighted in grey. **Bottom**: The mean absolute error between marker-based and markerless motion capture across the 30 participants at 2.5 m.s^−1^ (black) and 3.6 m.s^−1^ (grey).

**Figure 4 sensors-25-00934-f004:**
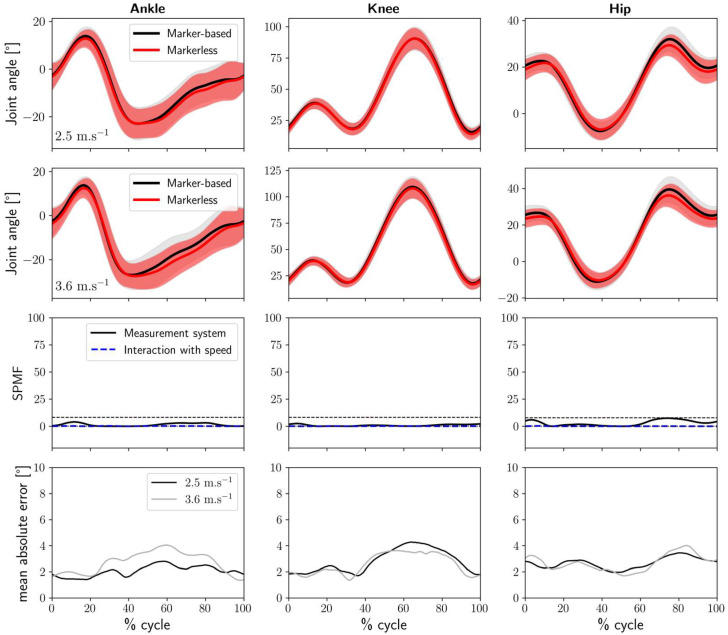
**Top**: Markerless motion capture sagittal plane joint angles at 2.5 m.s^−1^ and 3.6 m.s^−1^ for the ankle (**left column**), knee (**middle column**), and hip (**right column**) across the 30 participants for both clothing conditions. The solid lines represent the mean of the averaged angles of the 30 participants, and the shaded areas represent the standard deviation. **Middle**: F-statistics from SPM1D (SPMF) for the clothing condition (solid line) and the interaction of speed and clothing condition (dotted line), with dashed lines indicating the critical thresholds. **Bottom**: The mean absolute error between marker-based and markerless motion capture across the 30 participants at 2.5 m.s^−1^ (black) and 3.6 m.s^−1^ (grey).

**Figure 5 sensors-25-00934-f005:**
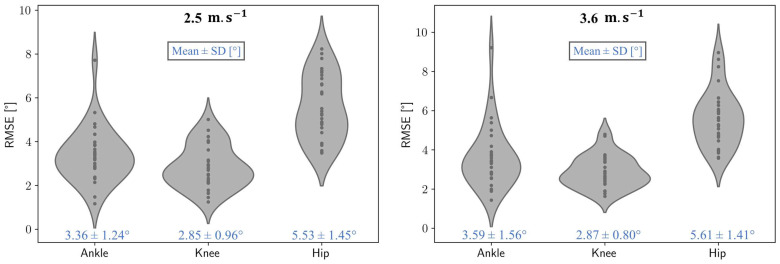
A violin plot showing the root mean square differences between the marker-based and markerless systems for the 30 participants at 2.5 m.s^−1^ (**left**) and 3.6 m.s^−1^ (**right**) for the ankle, knee, and hip joints. The mean values and associated standard deviations for each distribution are displayed below the plots.

**Table 1 sensors-25-00934-t001:** Mean RMSE ± standard deviation [°] per joint averaged over all percentage points of running cycle and participants at 2.5 m.s^−1^ and 3.6 m.s^−1^.

Speed	2.5 m.s^−1^	3.6 m.s^−1^
**Marker-based vs. markerless in Lab clothing condition**		
Ankle	3.36 ± 1.24°	3.59 ± 1.56°
Knee	2.85 ± 0.96°	2.87 ± 0.80°
Hip	5.53 ± 1.45°	5.61 ± 1.41°
**Markerless in Lab clothing vs. Running clothes conditions**		
Ankle	2.06 ± 0.88°	2.72 ± 1.81°
Knee	2.80 ± 1.75°	2.58 ± 1.42°
Hip	2.66 ± 1.35°	2.68 ± 1.27°

## Data Availability

The raw data supporting the conclusions of this article will be made available by the authors on request.

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
