# Peer review of "Impact of Running Clothes on Accuracy of Smartphone-Based 2D Joint Kinematic Assessment During Treadmill Running Using OpenPifPaf"

_sensors, 2025, doi:10.3390/s25030934_

Round 1
Reviewer 1 Report
Comments and Suggestions for Authors
Thank you for interesting paper. It was It was nice to read it. However, in my oppinion some methodological changes are necessary, as well as the title of the paper should be more realistic.
Please find some remarks in the attached file.

Author Response
Dear Reviewer,
We sincerely thank you for your thorough and constructive feedback on our manuscript. Your detailed comments and suggestions have provided valuable insights, enabling us to enhance the clarity, rigor, and overall quality of the paper.
We have carefully considered each of your remarks and made the necessary revisions to address your concerns. Specifically, we have clarified the methodologies for joint angle calculations, ensured consistency between marker-based and markerless analyses, and provided additional data and explanations to support the presented results.
Below, we provide a point-by-point response to each of your comments, detailing the changes made to the manuscript or providing clarifications where appropriate. We hope these revisions satisfactorily address your concerns and further strengthen the manuscript.
Thank you once again for your thoughtful review and for helping us improve our work.
Comments 1: Methodology of particular joint angle calculation should be presented clearly, in the reliable way:
a. For marker-based motion capture: The description presented in lines 122-128 and 144-146 (mentioned below) remains unclear, taking into account the usage of traditional set of markers (Plug-in Gait Lower-body' model with an additional marker placed on the middle of the lateral edge of each acromion).
lines 122-128: The approximate position of the left hip joint was estimated using the Newington-Gage model [26]. This estimation involved utilizing data from the four superior iliac spines as well as the participant's leg length. The 2D coordinates of markers placed on the participant's left side on the second metatarsal, lateral malleolus, lateral epicondyle of the femur, and shoulder, along with the calculated coordinates of the left hip joint center, will be used to determine the joint kinematics of the ankle, knee, and hip (see Figure 1).
lines 144-146: The 2D joint kinematics for the ankle, knee, and hip joints based on both the markers and keypoints coordinates were computed using trigonometry (see Figure 2B);
b. For markerless analysis: there is no reliable description of joint angle calculation methodology at all.
Response 1: Thank you for highlighting the need to clarify the methodology for calculating joint angles to ensure the reader fully understands the process. In light of comments 1 and 2, we believe two elements may have contributed to potential misunderstandings regarding the marker-based kinematics used as a comparison:
- Our primary objective, as mentioned at the end of the introduction, states that we are comparing the results "with those obtained using a marker-based reference motion analysis system." We recognize that this phrasing might be misleading and have reworded the sentence to better reflect our actual approach (line 93).
- While we noted that the markers were placed "to comply with the Nexus Plug-in Gait Lower-body model," this model was not used to calculate kinematics. Instead, the optoelectronic system was employed to extract accurate spatial coordinates of markers placed on anatomical landmarks, similar to the keypoints identified by OpenPifPaf. Joint kinematics were then calculated in a very similar manner for both methods using trigonometry, through the use of the dot product.
Ankle: The foot was considered as a rigid body. For the markerless method, the joint angle was calculated using the keypoints of the knee, ankle, and big toe. For the marker-based system, we used 2D coordinates of the markers placed on the lateral epicondyle, lateral malleolus, and second metatarsal. Ankle joint angles were corrected by accounting for the average angles measured in the neutral position during static calibration.
Knee: For the markerless method, the joint angle was computed using coordinates of keypoints for the hip, knee, and ankle. For the marker-based system, we used 2D coordinates of markers on the lateral malleolus, lateral epicondyle, and the hip position calculated using the Newington-Gage model, which is also employed in the Plug-in-Gait model.
Hip: For the markerless method, the joint angle was calculated using keypoints from the shoulder, hip, and knee. For the marker-based system, we used markers placed on the acromion and lateral epicondyle as well as the calculated 2D hip location.
We have revised the text in sections ‘2.3. Data Processing’ (lines 146-149;154-156;160;163-172) to clarify these points.
Note that we could have compared the results of the markerless method with those obtained in the sagittal plane by the Plug-in-Gait Lower-body model for the knee and ankle. However, the methodologies used for obtaining joint kinematics are quite different. Such a comparison would not be meaningful for the hip, as the two methods measure different parameters (this is addressed further in the next point of your review).
To improve clarity, we have also added a brief explanation of this distinction in the discussion (lines 315-323).
Comments 2: Taking into account available information (please take into account the remark 1) it is highly probable, that the definition of hip joint angle could be different in marker-based and markerless analyses. Please especially explain in Methodology, if pelvis movement was taken into account in the same way in both methods of analysis.
It seems, that Authors know, that such differences are present (Discussion, lines 308-309). Then, what is the sense of quantitative analysis of completely different kinematic parameters?
Response 2: We completely agree that it would not make sense to compare joint kinematics defined between the pelvis and the thigh with those defined between the trunk and the thigh. The first definition is the standard used in biomechanics, while the second is often applied clinically in scenarios where pelvic movement cannot be measured, such as with visual observation, goniometry, or certain movement analysis methods like Kinovea.
Although comparisons of such differing definitions ("apples and pears") can be found in the literature, this is absolutely not the case in our study. We hope that the explanation provided in point 1 regarding the calculation of joint angles has resolved any concerns about the validity of our comparisons.
Comments 3: Looking into Fig 2 b it is difficult to believe that the final results presented in Fig. 3a represent real data. Please provide some more detailed presentation of the source kinematic data to eliminate such doubts. Additionally, the kinematic results obtained for v=2.5 m s-1 as well as for wearing usual running clothes were not presented at all. Please present all discussed joint angles, before their statistical analyses.
Response 3: We apologize for any confusion caused by the explanation of the calculation method, which may have led to scepticism about the integrity of the data. We would like to clarify the steps taken to obtain the average kinematic curves (illustrated by figure 2) and subsequent results:
- Cycle Identification and Segmentation:
Figure 2A illustrates the process of determining the trigger movement to consistently select the same cycles (top) and the points where the signal was segmented into individual cycles (bottom). - Joint Kinematics Calculation:
Figure 2B shows the joint kinematics for the ankle, knee, and hip for the 30 seconds of motion capture. - Cycle Extraction and Averaging:
Using the data from Figure 2A (bottom), we extracted 20 cycles from the signal in Figure 2B and resampled them from frames to % cycle (shown as the grey lines in Figure 2C). We then calculated the average of the 20 cycles for each percentage of the cycle (represented by the black line in Figure 2C).
These average curves were computed for each of the 30 participants in each of the six conditions (marker-based, markerless in the lab clothing condition, and markerless in the running clothes condition each at two speeds). These average curves were used for calculating the RMSE, absolute errors, and performing the ANOVA analysis.
We have clarified the purpose of Figure 2 in the text to provide a clearer explanation for the reader (lines 173-177).
Figures 3 and 4 summarize the results of ANOVA analysis and absolute errors. The top panel of Figure 3 displays the mean values of the average curves across 30 participants and their standard deviations for the two measurement systems (marker-based and markerless) at one speed. Figure 4 follows the same format but shows results for the two clothing conditions.
Initially, both speeds were included in these graphs, but this made them difficult to interpret as the curves were very similar. To address this issue and improve transparency, we have modified Figures 3 and 4 (line 229-237 and 258-266) to include two lines representing the results at both speeds. We agree that this revision makes the presentation clearer and more transparent for the reader.
If there are still concerns regarding our data, we are willing to share it privately, subject to approval from the university, as the data is their property. Please let us know if this is deemed necessary.
Comments 4: Content of the Fig 3b was not explained in the Methodology.
Response 4: Thank you for raising this point. Fig 3b and Fig 4b are the results of the one-dimensional Statistical Parametric Mapping (SPM1D) two-way repeated-measures ANOVA. We have added an explanation in the point "2.4. Statistical analysis" to clarify the content presented in Fig 3b and Fig 4b (lines 219-224).
Comments 5: Authors used in the analysis references to foot-ground contact (lines 192-203). Please explain, what methodology was used to distinguish stance and swing phases during running.
Response 5: When dividing the cycles, we assumed that foot-ground contact occurred when the marker on the lateral malleolus of the left foot (for the marker-based system) or the keypoint corresponding to the left ankle (for the markerless system) reached its maximum forward position. This approach is illustrated in Figure 2.A., which shows the division of cycles based on the anterior-posterior location of the ankle marker. The "x's" in the figure represent the points where the signal is segmented to define each cycle.
It is important to note that we did not differentiate between the stance and swing phases in our analysis. Instead, we relied on Novacheck’s work on running kinematics [1] to interpret and explain the overall kinematic curves presented in our study. This clarification has been added to the text (lines 245-249).
Comments 6: The phrase clothing effects , especially in the title of the paper, is misleading. In reality, analysis of the clothing effect was reduced only to running clothes typically shorts or leggings (lines 102-103) and the difference between 'Lab clothing' and 'Running clothes' was probably quite small. If you want to present clothing effects , trousers or long dress should be taken into analyses.
Response 6: We completely agree with this observation. The studies we referenced also used the broad term "clothing effects" to describe experiments conducted with garments that were either close-fitting or specifically designed for running. It is also highly likely that the use of very loose-fitting clothing or dresses would negatively impact the performance of the markerless method. To better align with the content of our article, we have revised the title.
Reference:
- Novacheck, T.F. The Biomechanics of Running. Gait and Posture 1998.
Reviewer 2 Report
Comments and Suggestions for Authors
The overall quality of the article is high, and the research content is innovative and practical, providing new perspectives and methods for the field of running kinematics evaluation. Through rigorous experimental design and data analysis, valuable conclusions have been drawn, which have a positive impact on research and practice in related fields. Suggest further refining the details in subsequent revisions to make the article more comprehensive and persuasive.
The specific opinions are as follows:
1. It is suggested to further supplement the latest achievements of other researchers in the evaluation of running kinematics in recent years, as well as comparative analysis of different technical methods, in order to more comprehensively demonstrate the breakthroughs and contributions of this study on the basis of existing research.
2. Suggest adding specific processing procedures for the OpenPifPaf method in Materials and Methods.
3. When describing the experimental conditions, it is possible to provide a more detailed explanation of the specific process of participants adapting and warming up on the treadmill, as well as how gait stability and consistency are ensured when running at different speeds.
4. In the discussion, further comparisons can be made between the results of this study and previous related research, and possible reasons can be analyzed, such as differences in algorithms, data collection equipment, sample characteristics, and other aspects used in different studies. In addition, for the prospects of future research directions, more specific innovative research ideas or technological improvement plans can be proposed to promote further development in this field.
Author Response
Dear Reviewer,
We sincerely thank you for your positive and encouraging feedback regarding our manuscript. We are delighted to hear that you found the overall quality of the article to be high, and that our research was recognized as innovative and practical, with meaningful contributions to the field of running kinematics evaluation.
We greatly appreciate your suggestions for refining the details to further enhance the comprehensiveness and persuasiveness of the article. In response, we have carefully reviewed your specific comments and implemented revisions to address them. We believe these changes improve the clarity, depth, and overall quality of the manuscript.
Below, we provide detailed responses to each of your suggestions and outline the corresponding modifications made to the manuscript.
Comments 1: It is suggested to further supplement the latest achievements of other researchers in the evaluation of running kinematics in recent years, as well as comparative analysis of different technical methods, in order to more comprehensively demonstrate the breakthroughs and contributions of this study on the basis of existing research.
Response 1: In response to your suggestion, we have expanded the introduction to include an explanation of various current methods used for markerless analysis (lines 57–60). Furthermore, we have detailed approaches for obtaining joint kinematics of running in the sagittal plane using computer vision and cited studies that have performed 2D measurements of running kinematics in the sagittal plane using smartphone video (lines 66–77).
We hope these additions clarify the scope and objectives of our study, which aim to evaluate the accuracy of a simple, accessible, and user-friendly system based on OpenPifPaf for assessing joint kinematics in the sagittal plane from smartphone video.
Comments 2: Suggest adding specific processing procedures for the OpenPifPaf method in Materials and Methods.
Response 2: As requested, we have added specific details regarding the processing procedures for the OpenPifPaf method in Section “2.3. Data processing.” (lines 163-177).
Comments 3: When describing the experimental conditions, it is possible to provide a more detailed explanation of the specific process of participants adapting and warming up on the treadmill, as well as how gait stability and consistency are ensured when running at different speeds.
Response 3: We have added additional details to Section “2.2. Procedure” (lines 123-134) to provide a more comprehensive explanation of the specific process by which participants adapt and warm up on the treadmill, as well as the measures taken to ensure gait stability and consistency when running at different speeds.
Comments 4: In the discussion, further comparisons can be made between the results of this study and previous related research, and possible reasons can be analyzed, such as differences in algorithms, data collection equipment, sample characteristics, and other aspects used in different studies. In addition, for the prospects of future research directions, more specific innovative research ideas or technological improvement plans can be proposed to promote further development in this field.
Response 4: We have added further details to the discussion section regarding the comparison with other studies (lines 313-321;325-327).
The next step in this study is to apply the developed method to measure running kinematics in ACL patients within clinical practices. Additionally, another project aims to use this method for measurements in Benin, where access to motion analysis laboratories is currently unavailable. Details regarding these future directions have been included before the conclusion (lines 372-376).

Round 2
Reviewer 1 Report
Comments and Suggestions for Authors
Thank you for all changes in the paper.
Reviewer 2 Report
Comments and Suggestions for Authors
I don't have any further questions.
Comments on the Quality of English Languagefine